

# AMAY-Net: adaptive multi-scale attention YOLO network for liver and gallbladder segmentation in laparoscopic cholecystectomy

Yuyang Zhou[1], Yulai You[2], Xiaokai Tan[2] and Juncheng Tang[2]

[1] Department of Hepatobiliary Pancreatic Surgery, People's Hospital of Longhua, Shenzhen, Shenzhen, China
[2] Department of Hepatobiliary Surgery, Chongqing University Jiangjin Hospital, Chongqing, Chongqing, China

## ABSTRACT

This article introduces a novel liver and gallbladder segmentation framework, named Adaptive Multi-Scale Attention YOLO Network (AMAY-Net), designed for semantic segmentation of laparoscopic cholecystectomy images. Building upon the powerful feature extraction capabilities of You Only Look Once (YOLO), AMAY-Net incorporates several advanced modules to enhance performance in medical image segmentation tasks. First, a multi-scale feature extraction module is employed to capture anatomical structures of various sizes, ensuring effective detection of large organs like the liver and smaller structures such as the gallbladder and surgical instruments. Second, an adaptive class-balancing loss function is implemented to dynamically adjust the weights of underrepresented classes, improving the segmentation accuracy of small structures. Additionally, the network integrates a spatial and channel attention mechanism, enhancing the focus on critical regions in the image. Finally, residual connections are introduced in the YOLO backbone to improve feature propagation and gradient flow efficiency. Experimental results demonstrate that AMAY-Net achieves superior performance on the CholecSeg8k dataset, with significant improvements in the segmentation accuracy of key anatomical structures such as the liver and gallbladder.

## INTRODUCTION

Laparoscopic cholecystectomy, a minimally invasive surgery for gallbladder removal, is one of the most common procedures worldwide (*Agarwal et al., 2015*; *Zimmitti et al., 2016*; *Ye et al., 2015*; *Agresta et al., 2015*). Accurate segmentation of critical anatomical structures such as the liver, gallbladder, cystic duct, and hepatic veins can provide valuable intraoperative visual guidance, potentially assisting surgeons in decision-making and helping reduce procedural complexity (*Madani et al., 2022*; *Mascagni et al., 2022*; *Tang et al., 2018*). However, traditional image processing methods face challenges due to

Corresponding author
Juncheng Tang,
tangjunchengchina@126.com

variability in organ shapes, occlusions caused by surgical instruments, and inconsistent imaging conditions during surgery.

Deep learning-based methods, particularly convolutional neural networks (CNNs), have made significant advances in medical image segmentation. Widely used architectures like U-Net (*Ronneberger, Fischer & Brox, 2015*; *Zunair & Hamza, 2021*) and Fully Convolutional Networks (FCN) (*Long, Shelhamer & Darrell, 2015*; *Sun & Wang, 2018*) have set benchmarks for pixel-level segmentation tasks due to their ability to capture fine details of anatomical structures. *Ronneberger, Fischer & Brox (2015)* proposed U-Net, which has gained widespread adoption because of its encoder-decoder structure and skip connections that help preserve spatial information at different scales. DeepLab (*Chen et al., 2017a*) and Mask R-CNN (*Chen et al., 2017b*) have further advanced segmentation with their ability to handle multi-scale objects and perform instance segmentation. Despite their accuracy, these models often struggle with real-time performance in dynamic environments like laparoscopic surgery, where speed is as crucial as segmentation precision.

On the other hand, You Only Look Once (YOLO) (*Redmon, 2016*; *Gallagher & Oughton, 2024*) is widely recognized for its efficiency in real-time object detection, predicting objects at multiple scales in a single forward pass. While YOLO is primarily designed for object detection, it lacks the pixel-level accuracy needed for segmentation tasks. This makes it less suitable for medical imaging tasks such as liver and gallbladder segmentation, which demand precise delineation of anatomical boundaries. In contrast, models like U-Net and DeepLab are computationally intensive and thus less suited for real-time applications (*Jiao et al., 2020*; *Xiao et al., 2020*). Consequently, there is a need for a model that balances both speed and accuracy, especially in surgical environments.

To address these challenges, we propose Adaptive Multi-Scale Attention YOLO Network (AMAY-Net), a framework that extends YOLO's real-time detection capabilities to medical image segmentation by integrating multi-scale feature extraction, attention mechanisms, and an adaptive class-balancing loss function. The objective is not to replace models like U-Net or DeepLab, but to offer a solution specifically optimized for the real-time demands of laparoscopic surgery while maintaining segmentation accuracy. Compared to U-Net, DeepLab, and Mask R-CNN, AMAY-Net provides several advantages in specific use cases. AMAY-Net leverages YOLO's speed, making it ideal for time-sensitive applications such as surgery, where real-time feedback is essential. Moreover, it introduces a multi-scale feature extraction module that captures both large and small structures, such as the liver and gallbladder, which is often a challenge for conventional segmentation models. Additionally, AMAY-Net addresses the issue of class imbalance—often found in medical datasets—by implementing an adaptive class-balancing loss function that ensures that smaller but clinically significant structures are not overlooked during training.

As illustrated in Fig. 1, AMAY-Net bridges the inherent trade-off between segmentation accuracy and inference speed. Traditional models like U-Net and DeepLab (left) achieve pixel-level precision but suffer from computational bottlenecks, while YOLO-based approaches (right) prioritize real-time performance at the cost of boundary delineation

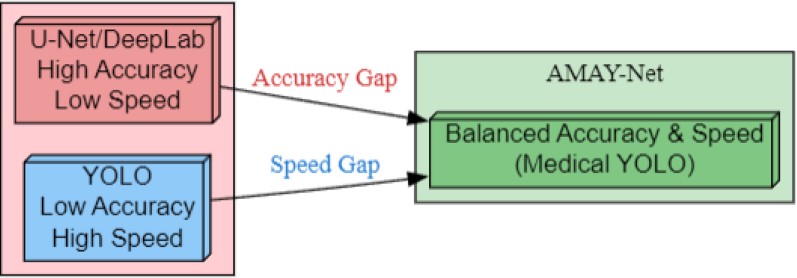

**Figure 1** **Bridging the accuracy-speed gap with AMAY-Net for medical image segmentation.**

fidelity. Our framework (center) introduces three critical enhancements: (1) multi-scale feature fusion for preserving anatomical details, (2) adaptive attention mechanisms to handle surgical occlusions, and (3) a class-balanced loss function optimized for gallbladder segmentation. This hybrid architecture enables sub-second inference while maintaining a high Intersection over Union (IoU) on critical structures, fulfilling the dual requirements of laparoscopic guidance systems.

## RELATED WORK

Semantic segmentation has become widely utilized in medical imaging to effectively identify and delineate anatomical structures from complex, high-resolution images, providing valuable assistance in clinical interpretation and treatment planning. Early approaches to medical image segmentation relied heavily on region-based methods and classical machine learning techniques. These methods, including region-growing algorithms and graph-based approaches, often struggled with the variability and complexity of medical images, especially in scenarios involving soft tissues or structures with vague boundaries. While transformer-based architectures like SegFormer (*Xie et al., 2021*) demonstrate efficient multi-scale modeling through hierarchical attention, their computational demands remain prohibitive for real-time surgical applications requiring sub-second latency. While these traditional methods offered some degree of success, they were largely dependent on hand-crafted features and lacked the robustness needed for large-scale clinical use.

The advent of deep learning, particularly convolutional neural networks (CNNs), marked a significant breakthrough in medical image segmentation. Models such as Fully Convolutional Networks (FCN) (*Long, Shelhamer & Darrell, 2015*) and U-Net (*Alshomrani, Arif & Al Ghamdi, 2023*) introduced the concept of end-to-end learning for dense pixel prediction, allowing networks to learn hierarchical representations directly from data. *Alshomrani, Arif & Al Ghamdi (2023)* proposed U-Net, characterized by its symmetric encoder-decoder structure and skip connections, which quickly became the preferred architecture for many medical imaging tasks, particularly organ segmentation. Leveraging both local and global contexts through skip connections, U-Net demonstrated superior performance compared to traditional methods. Variants such as Attention U-Net (*Oktay et al., 2018*), Res-UNet (*Zhou et al., 2021*; *Alom et al., 2019*), and Swin U-Net

(*Cao et al., 2022*) enhanced segmentation performance by incorporating attention mechanisms and residual learning, respectively. However, while these models excel in terms of segmentation accuracy, they often suffer from computational inefficiencies, making them less suitable for real-time applications like laparoscopic surgery.

In parallel, object detection models such as YOLO (*Redmon, 2016*) gained prominence for their ability to efficiently detect objects in real-time, achieving impressive speeds by predicting bounding boxes for multiple objects in a single forward pass. YOLO's capability to handle multi-scale objects and perform real-time inference has led to its exploration in medical image analysis, particularly for tasks requiring rapid feedback, such as surgical navigation (*Ragab et al., 2024*; *Soni & Rai, 2024*). However, YOLO's design inherently suits bounding box prediction rather than dense pixel-wise segmentation, limiting its direct application to tasks like organ segmentation.

Multi-scale feature extraction has been recognized as critical for handling the diverse sizes of anatomical structures in medical images. *Lin et al. (2017)* proposed Feature Pyramid Networks (FPN), introducing a hierarchical structure that aggregates features from multiple resolutions to capture both large and small structures. In the context of medical segmentation, DeepLabV3+ (*Chen et al., 2018*; *Gopikrishna et al., 2024*) applied atrous convolutions to enlarge the receptive field while maintaining fine resolution, enhancing the ability to capture complex structures like tumors or lesions across varying scales. Yet, while these methods address the issue of multi-scale object handling, their computational complexity still poses challenges for real-time clinical environments. *Kolbinger et al. (2023)* proposed using machine learning methods, specifically DeepLabv3 and SegFormer architectures, for anatomical structure segmentation in laparoscopic surgery. They compared the performance of these models with human experts in pancreas segmentation, finding that the machine learning models outperformed most human participants, demonstrating their potential for real-time clinical assistance.

A common challenge in medical image segmentation is class imbalance, where larger structures dominate the image while smaller yet clinically important structures (*e.g.*, small lesions or ducts) are underrepresented. Recent efforts like the Dresden Surgical Anatomy Dataset (*Carstens et al., 2023*) have further highlighted the challenges of organ segmentation in laparoscopic environments, particularly in scenarios involving dynamic instrument interaction and tissue deformation. Techniques such as weighted cross-entropy and focal loss (*Lin, 2017*; *Ağralı & Kılıç, 2024*) have been developed to address this issue by assigning higher weights to underrepresented classes, ensuring that the network focuses on these harder-to-classify regions. *Sudre et al. (2017)* introduced Generalized Dice Loss, specifically targeting class imbalance by normalizing each class's contribution, thereby improving segmentation of smaller structures. Despite these advancements, balancing the focus on both small and large anatomical structures efficiently remains an ongoing research challenge. Attention mechanisms have also gained traction in improving segmentation performance by guiding the model to focus on the most relevant regions of an image. *Oktay et al. (2018)* introduced attention gates in Attention U-Net to dynamically highlight important regions, while recent models like the Convolutional Block Attention Module (CBAM) (*Woo et al., 2018*) combine spatial and

channel attention to further refine the network's focus on meaningful features. These mechanisms have significantly improved segmentation accuracy, particularly in tasks involving complex structures with overlapping regions.

While each of these advancements has addressed key challenges in medical image segmentation—multi-scale feature handling, class imbalance, and attention mechanisms—there remains a gap in models offering both high segmentation accuracy and real-time performance, particularly in dynamic surgical environments like laparoscopic procedures. To address this, we propose AMAY-Net, a framework that integrates multi-scale feature extraction, adaptive class-balancing loss, and dual attention mechanisms into the efficient YOLO architecture. This novel approach enables precise liver and gallbladder segmentation while maintaining real-time inference speed, crucial for intraoperative guidance and decision-making in laparoscopic surgery.

## METHODOLOGY

As shown in Fig. 2, the proposed architecture, AMAY-Net, first processes the input image through a YOLO-based backbone for initial feature extraction. The extracted features are then passed through both downsample and upsample layers to capture multi-scale information, with skip connections to preserve spatial details from earlier layers. Following this, dual attention mechanisms (spatial and channel attention modules) are applied separately to refine spatial and channel-wise features, and their outputs are combined *via* an element-wise addition for feature enhancement. Finally, a dedicated segmentation head generates the segmentation mask, while a parallel detection head produces bounding box coordinates (x, y, w, h), objectness scores, class predictions, and IoU scores, leading to the final segmentation and detection outputs.

### YOLO backbone and multi-scale feature extraction

The input image, denoted as $\mathbf{I} \in \mathbb{R}^{H \times W \times C}$, where $H$, $W$, and $C$ represent the height, width, and number of channels, is first processed by the YOLO backbone for initial feature extraction. This process generates a feature map $\mathbf{F}_{yolo}$, which serves as the foundation for further multi-scale processing:

$$\mathbf{F}_{yolo} = \text{YOLO}(\mathbf{I}). \tag{1}$$

To capture higher-level semantic information, we employ downsampling layers that progressively reduce the spatial resolution of the feature maps. The first downsampling operation generates a feature map $\mathbf{F}_{down1}$ from the output of the YOLO backbone:

$$\mathbf{F}_{down1} = \text{Downsample}_1(\mathbf{F}_{yolo}). \tag{2}$$

This step typically involves convolutional layers with a stride of 2 or max-pooling, which reduces the spatial dimensions of the feature map while increasing the receptive field. The second downsampling operation further reduces the resolution, producing $\mathbf{F}_{down2}$:

$$\mathbf{F}_{down2} = \text{Downsample}_2(\mathbf{F}_{down1}). \tag{3}$$

At this stage, the model captures even more abstract features, which are useful for detecting larger structures within the image, but at the cost of spatial resolution. To recover fine

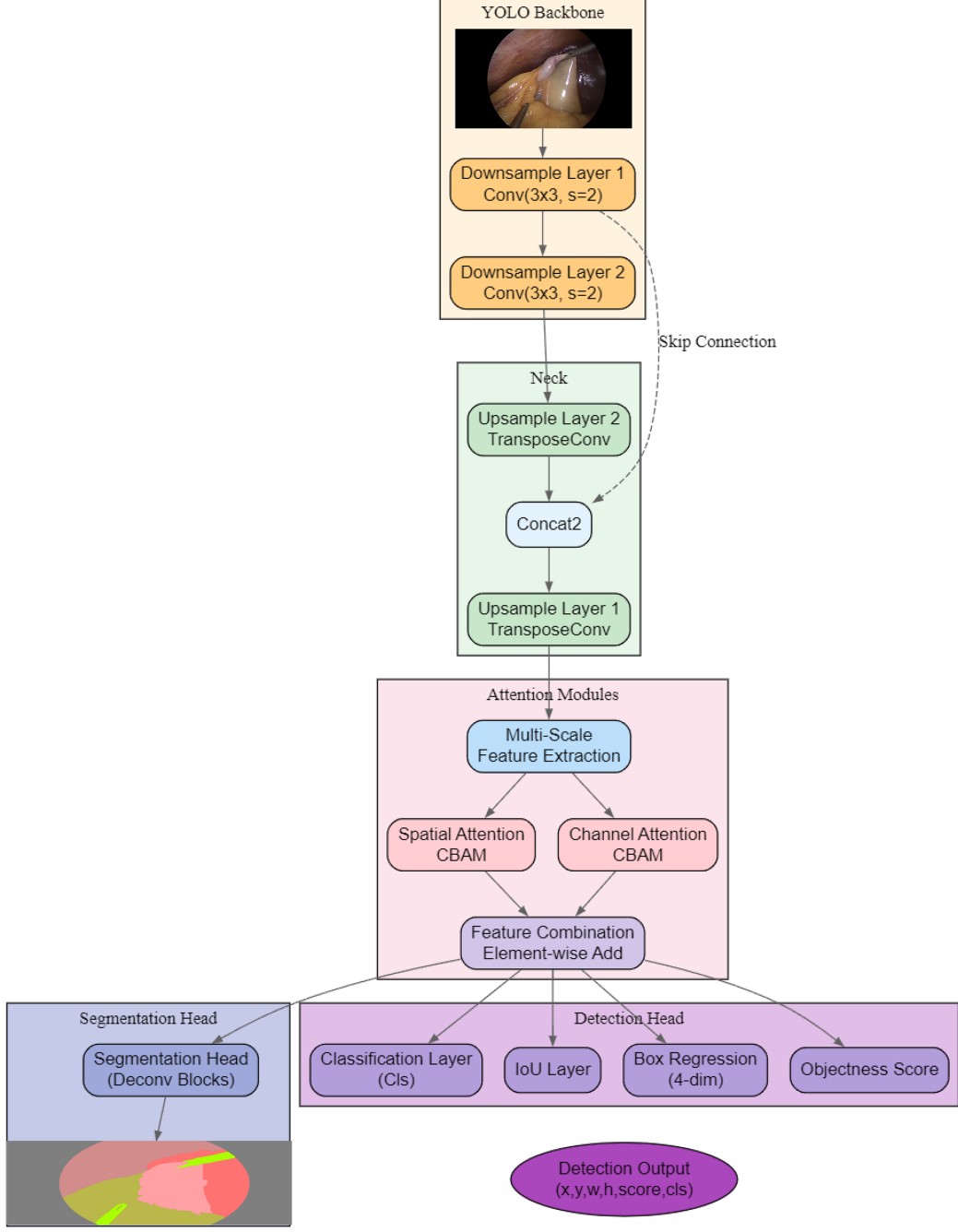

**Figure 2 Overview of the proposed AMAY-Net architecture, integrating multi-scale feature extraction, dual attention mechanisms, and parallel segmentation and detection heads.**

details that may have been lost during downsampling, the network employs upsampling layers. The first upsampling layer, Upsample$_1$, takes $\mathbf{F}_{down1}$ as input and increases the spatial resolution:

$$\mathbf{F}_{up1} = \text{Upsample}_1(\mathbf{F}_{down1}). \tag{4}$$

Similarly, the second upsampling layer, Upsample$_2$, operates on $\mathbf{F}_{down2}$:

$$\mathbf{F}_{up2} = \text{Upsample}_2(\mathbf{F}_{down2}). \tag{5}$$

These upsampling operations typically use transposed convolutions or bilinear interpolation to restore the spatial resolution, allowing the model to focus on smaller anatomical structures. Finally, the output feature maps from both the downsample and upsample layers are concatenated along with the original $\mathbf{F}_{yolo}$ feature map in the multi-scale feature extraction module:

$$\mathbf{F}_{multi} = \text{Concat}(\mathbf{F}_{yolo}, \mathbf{F}_{down1}, \mathbf{F}_{down2}, \mathbf{F}_{up1}, \mathbf{F}_{up2}). \tag{6}$$

This concatenated feature map $\mathbf{F}_{multi}$ captures information across multiple scales, enabling the network to effectively handle both small and large anatomical structures in the image. To prepare these features for the next stage (attention mechanisms), the concatenated feature map is processed through a $1 \times 1$ convolution to reduce its channel dimensionality:

$$\mathbf{F}_{final} = \text{Conv}_{1\times1}(\mathbf{F}_{multi}). \tag{7}$$

This resulting feature map $\mathbf{F}_{final}$ serves as the input to the subsequent attention modules.

## Attention mechanisms

After the multi-scale feature extraction process, where the fused feature map $\mathbf{F}_{final}$ is produced, the network applies two sequential attention mechanisms to further refine these features: Spatial Attention and Channel Attention. These attention mechanisms help the network focus on the most relevant regions and feature channels in the image, enhancing the quality of the segmentation. The spatial attention mechanism is designed to help the network focus on important regions of the image by adjusting the spatial importance of different areas. Given the multi-scale feature map $\mathbf{F}_{final} \in \mathbb{R}^{H \times W \times C}$, the spatial attention module generates a spatial attention map $\mathbf{A}_{spatial} \in \mathbb{R}^{H \times W}$, which highlights the regions that are most relevant to the task, such as regions containing organs or other key anatomical structures. The spatial attention mechanism is typically computed using the average and maximum pooling operations along the channel dimension. These pooling operations summarize the channel-wise information into two separate spatial maps, which are then concatenated and passed through a convolutional layer to produce the final attention map $\mathbf{A}_{spatial}$:

$$\mathbf{A}_{spatial} = \sigma\big(\text{Conv}_{7\times7}\big([\text{AvgPool}(\mathbf{F}_{final}), \text{MaxPool}(\mathbf{F}_{final})]\big)\big). \tag{8}$$

Here, $\sigma$ represents the sigmoid activation function, and $[\cdot, \cdot]$ denotes concatenation along the channel dimension. The convolutional layer uses a $7 \times 7$ kernel to capture local spatial relationships.

The attention map $\mathbf{A}_{spatial}$ is then element-wise multiplied with the input feature map $\mathbf{F}_{final}$ to generate the spatially refined feature map $\mathbf{F}_{spatial}$:

$$\mathbf{F}_{spatial} = \mathbf{A}_{spatial} \odot \mathbf{F}_{final}. \tag{9}$$

This operation ensures that the network emphasizes the regions of the image that are most relevant to the segmentation task, suppressing less important areas.

After spatial attention refines the feature map at the spatial level, the next step is to apply channel attention, which focuses on the importance of each feature channel. Channel attention adjusts the weight of each channel based on its relevance to the segmentation task. To compute the channel attention, the feature map $\mathbf{F}_{spatial}$ is first compressed along the spatial dimensions using global average pooling and global max pooling, resulting in two vectors that represent the global context across all channels:

$$\mathbf{V}_{avg} = \text{GlobalAvgPool}(\mathbf{F}_{spatial}), \quad \mathbf{V}_{max} = \text{GlobalMaxPool}(\mathbf{F}_{spatial}). \tag{10}$$

Both $\mathbf{V}_{avg}$ and $\mathbf{V}_{max}$ are vectors of size $C$, where $C$ is the number of channels in the feature map. These vectors are then passed through a shared two-layer fully connected network to compute the channel attention scores:

$$\mathbf{A}_{channel} = \sigma\big(\text{FC}_2\big(\text{ReLU}\big(\text{FC}_1\big(\mathbf{V}_{avg} + \mathbf{V}_{max}\big)\big)\big)\big). \tag{11}$$

In this equation, $\sigma$ represents the sigmoid activation function, and $\text{FC}_1$ and $\text{FC}_2$ are fully connected layers that reduce and then restore the dimensionality of the channel vectors. The resulting attention map $\mathbf{A}_{channel} \in \mathbb{R}^C$ contains a weight for each channel, indicating its importance. The channel attention map $\mathbf{A}_{channel}$ is then applied to the spatially refined feature map $\mathbf{F}_{spatial}$ *via* element-wise multiplication:

$$\mathbf{F}_{channel} = \mathbf{A}_{channel} \odot \mathbf{F}_{spatial}. \tag{12}$$

This operation adjusts the contribution of each feature channel, allowing the network to emphasize the most important features for segmentation. The output of the channel attention mechanism, $\mathbf{F}_{channel}$, is the final refined feature map that incorporates both spatial and channel-wise attention. This refined feature map is then passed to the next stage of the network, the segmentation head, where it is further processed to produce the final segmentation output. By sequentially applying spatial and channel attention, the network ensures that it not only focuses on the most relevant regions of the image but also prioritizes the most informative feature channels. This two-step attention process enhances the model's ability to capture important anatomical structures, leading to more accurate segmentation results.

## Segmentation head and final output

After applying the dual attention mechanisms, the refined feature map $\mathbf{F}_{channel}$, which incorporates both spatial and channel-wise information, is passed to the segmentation head. The segmentation head is responsible for converting these refined features into meaningful outputs for object detection and segmentation. Specifically, the segmentation head consists of several key components, each focusing on a distinct aspect of the task. First, the classification layer is responsible for classifying the detected regions in the feature map into predefined categories. Let $\mathbf{F}_{cls}$ be the input feature map to this layer, which is derived from the output of the channel attention mechanism $\mathbf{F}_{channel}$. The classification layer applies a convolutional operation followed by a softmax function to generate a probability distribution over $K$ object categories for each pixel:

$$\mathbf{P}_{cls}(x, y, k) = \text{softmax}(\text{Conv}_{cls}(\mathbf{F})), \quad k \in \{1, 2, \ldots, K\}. \tag{13}$$

Here, $\mathbf{P}_{cls}(x, y, k)$ represents the probability that the pixel at position $(x, y)$ belongs to class $k$. The classification output provides the likelihood of each pixel being associated with a specific object category, aiding in both object detection and semantic segmentation.

Next, the IoU layer is responsible for calculating the Intersection* over Union (IoU) between the predicted segmentation masks and the ground truth masks. The IoU is a common metric used to measure the accuracy of object localization and segmentation by comparing the overlap between the predicted and true bounding boxes or masks. Given a predicted mask $\mathbf{M}_{pred}$ and a ground truth mask $\mathbf{M}_{gt}$, the IoU is computed as follows:

$$\text{IoU} = \frac{\mathbf{M}_{pred} \cap \mathbf{M}_{gt}}{\mathbf{M}_{pred} \cup \mathbf{M}_{gt}}. \tag{14}$$

The IoU score provides a measure of how well the predicted regions align with the true regions, with values ranging from 0 (no overlap) to 1 (perfect overlap). The IoU layer outputs this score for each detected object, which is then used during training to optimize the segmentation performance.

The box regression layer refines the bounding box predictions by adjusting their position and size. This layer takes the output feature map $\mathbf{F}_{box}$ from the segmentation head and predicts a set of bounding box coordinates for each detected object. The box regression process can be formulated as a regression problem, where the goal is to minimize the difference between the predicted and ground truth bounding box coordinates. Let $\mathbf{B}_{pred}$ represent the predicted bounding box coordinates, and $\mathbf{B}_{gt}$ represent the ground truth bounding box coordinates. The objective of the box regression layer is to minimize the following loss function:

$$\mathscr{L}_{box} = \sum_{i=1}^{4} \left( \mathbf{B}_{pred}^{i} - \mathbf{B}_{gt}^{i} \right)^2. \tag{15}$$

This loss function penalizes the differences between the predicted and ground truth bounding box coordinates, thereby refining the predicted bounding boxes to ensure they tightly fit around the detected objects.

The objectness score layer is responsible for determining whether a given region contains an object or not. This layer outputs a score, $\mathbf{S}_{obj}(x, y)$, for each region of interest, which indicates the likelihood that an object is present at a particular location in the image:

$$\mathbf{S}_{obj}(x, y) = \sigma \big( \text{Conv}_{obj}(\mathbf{F}_{obj}) \big). \tag{16}$$

Here, $\sigma$ represents the sigmoid activation function, and $\mathbf{F}_{obj}$ is the input feature map for this layer. The objectness score layer helps the network distinguish between regions that contain objects and those that belong to the background. This is crucial for both object detection and segmentation, as it filters out regions that are not relevant for the task.

One of the primary challenges in medical image segmentation is dealing with the imbalance between classes, especially in datasets like CholecSeg8k, where larger structures such as the liver dominate, while smaller structures like the gallbladder and surgical instruments are underrepresented. Standard loss functions, such as cross-entropy, often struggle with this imbalance, leading to poor performance on the underrepresented classes.

To address this issue, we introduce an adaptive class-balancing loss function that not only accounts for class imbalance but also dynamically adjusts the loss contributions based on the difficulty of classifying smaller structures. This is achieved by a combination of two mechanisms: classes that are underrepresented in the dataset are assigned higher weights during training, ensuring that they receive more attention. The weight for each class is inversely proportional to the frequency of that class in the dataset, allowing the network to focus more on difficult-to-learn classes. We further enhance the loss by integrating focal loss, which is particularly effective for addressing class imbalance. Focal loss places more emphasis on hard-to-classify examples, such as small structures or objects that are frequently misclassified.

The adaptive class-balancing loss function assigns different weights to the loss components for each class based on their frequency in the dataset. Classes that are underrepresented receive higher weights, while more common classes are given lower weights. This dynamic adjustment helps the model focus more on learning the less frequent classes, preventing them from being overshadowed by dominant classes during training. Let $\mathbf{w}_k$ represent the weight assigned to class $k$, and $N_k$ be the number of pixels (or instances) belonging to class $k$ in the training dataset. The weight for each class is computed as follows:

$$\mathbf{w}_k = \frac{1}{N_k + \varepsilon} \tag{17}$$

where $\varepsilon$ is a small constant to avoid division by zero. This formulation ensures that classes with fewer instances (*i.e.*, smaller structures) receive higher weights, encouraging the model to pay more attention to them during training.

In addition to class frequency weighting, we integrate the focal loss mechanism to focus more on hard-to-classify examples. Focal loss is designed to handle class imbalance by down-weighting easy examples and up-weighting hard examples. This is particularly useful when dealing with small anatomical structures or objects that are frequently occluded or hard to distinguish from the background.

The focal loss for class $k$ is defined as:

$$\mathscr{L}_{focal} = -\mathbf{w}_k(1 - \mathbf{P}_{cls}(x,y,k))^\gamma \log(\mathbf{P}_{cls}(x,y,k)) \tag{18}$$

where $\gamma$ is a focusing parameter that controls how much emphasis is placed on hard examples. When $\gamma$ is increased, the focal loss gives more weight to hard-to-classify examples, such as small or underrepresented structures.

The total loss $\mathscr{L}_{total}$ is composed of four key components: classification loss, IoU loss, box regression loss, and objectness score loss. Each of these components is scaled by a corresponding weight that adapts based on the class distribution and the difficulty of the examples:

$$\mathscr{L}_{total} = \lambda_{cls}\mathscr{L}_{cls} + \lambda_{IoU}\mathscr{L}_{IoU} + \lambda_{box}\mathscr{L}_{box} + \lambda_{obj}\mathscr{L}_{obj}. \tag{19}$$

Here, $\lambda_{cls}$, $\lambda_{IoU}$, $\lambda_{box}$, and $\lambda_{obj}$ are the adaptive weights for classification, IoU, box regression, and objectness score, respectively.

The classification loss is computed using a combination of weighted cross-entropy and focal loss to address class imbalance. This ensures that the model does not disproportionately focus on the dominant classes. The classification loss is defined as:

$$\mathscr{L}_{cls} = \sum_{k=1}^{K} \mathbf{w}_k \mathscr{L}_{focal} \tag{20}$$

where $\mathbf{P}_{cls}(x, y, k)$ is the predicted probability for class $k$ at pixel $(x, y)$, and $\mathscr{L}_{focal}$ is the focal loss for hard-to-classify examples.

The IoU loss penalizes low overlap between the predicted and ground truth segmentation masks. To maximize overlap, we define the IoU loss as:

$$\mathscr{L}_{IoU} = 1 - \text{IoU}. \tag{21}$$

This loss encourages the model to produce segmentation masks that closely match the ground truth, particularly for small or irregularly shaped structures.

The objectness score loss is a binary cross-entropy loss that measures the accuracy of predicting whether a region contains an object. It is defined as:

$$\mathscr{L}_{obj} = -\Big(\mathbf{y}_{obj} \log \mathbf{S}_{obj} + (1 - \mathbf{y}_{obj}) \log(1 - \mathbf{S}_{obj})\Big) \tag{22}$$

where $\mathbf{y}_{obj}$ is the ground truth label indicating whether an object is present, and $\mathbf{S}_{obj}$ is the predicted objectness score.

The outputs of the classification, IoU, box regression, and objectness score layers are combined to form the final segmentation output. Specifically, the final pixel-wise prediction for the input image is generated by combining the classification probabilities, refined bounding box coordinates, and objectness scores. The final segmentation output $\mathbf{S}_{seg}$ can be represented as:

$$\mathbf{S}_{seg}(x, y) = \mathbf{P}_{cls}(x, y, k) \cdot \mathbf{S}_{obj}(x, y). \tag{23}$$

This output represents the likelihood that each pixel belongs to a particular object class, and it is further refined by the objectness score, ensuring that only regions containing objects are considered in the final prediction. The bounding box information is also used to adjust the regions of interest for object detection. By integrating multiple layers within the segmentation head and incorporating the adaptive class-balancing loss function, AMAY-Net is capable of performing both object detection and segmentation with high accuracy. The classification layer assigns object categories to each region, the IoU layer evaluates the segmentation quality, the box regression layer refines the object boundaries, and the objectness score layer filters out irrelevant regions. Together, these components ensure that the final segmentation output is both accurate and precise, while addressing the challenges posed by class imbalance in medical image segmentation.

# EXPERIMENTAL RESULTS AND ANALYSIS

## Dataset

In this work, we utilize the CholecSeg8k dataset, which is specifically designed for semantic segmentation tasks in laparoscopic cholecystectomy. CholecSeg8k is based on the

widely-used Cholec80 dataset (*Hong et al., 2020*), a collection of 80 video recordings of laparoscopic cholecystectomy surgeries. From this dataset, 8,080 frames were extracted from 17 different surgical videos and annotated at the pixel level with 13 distinct classes. These classes include key anatomical structures, such as the liver, gallbladder, and gastrointestinal tract, as well as surgical instruments, like graspers and L-hook electrocautery devices.

CholecSeg8k provides detailed pixel-level annotations, making it an ideal resource for developing and evaluating semantic segmentation models in the context of minimally invasive surgery. The dataset contains images with a resolution of $854 \times 480$ pixels, and each image is accompanied by three types of masks: the color mask for visualization, the annotation mask for training, and the watershed mask for precise boundary delineation during algorithm development. To ensure rigorous model training and evaluation, the dataset was split strictly based on surgical cases, meaning that all images from a given surgical video were assigned entirely to a single subset. This approach prevents data leakage and ensures that the test set represents truly independent surgical procedures, which is critical for assessing generalization in clinical settings.

The training set consisted of 6,560 images collected from videos 01, 09, 12, 17, 18, 20, 24, 25, 26, 27, 28, 35, 37, 48, and 52. These cases were used exclusively for training the model. The validation set included 720 images from video 43 and was used for tuning hyperparameters and monitoring the model's performance during training. Finally, the test set comprised 800 images from video 55, which was completely held out during training and validation, and used solely for final performance evaluation.

## Experimental setup

All experiments were conducted on a system equipped with an Intel Xeon CPU, 64 GB of RAM, and an NVIDIA GTX 1660 GPU. All models, including our proposed AMAY-Net and comparison methods (DeepLabV3+, Attention U-Net, U-Net, YOLOv5, YOLOv10, and SegFormer), were implemented in PyTorch to effectively utilize GPU acceleration. To ensure reproducibility and fair comparisons, a consistent training protocol was maintained across all models.

Each model was trained using the Adam optimizer with an initial learning rate of 0.001, and the learning rate was uniformly reduced by a factor of 0.1 if the validation loss plateaued for five consecutive epochs. A batch size of 16 was applied during training, and early stopping was enforced if no improvement in validation performance was observed after 10 consecutive epochs, with the maximum number of training epochs capped at 100. All models used a combination of Dice loss and binary cross-entropy loss, except for AMAY-Net, which employed an adaptive class-balancing loss function to specifically address class imbalance. Additionally, dataset splits were strictly based on surgical cases to prevent data leakage, ensuring robust and fair performance evaluation across methods.

For model-specific configurations, DeepLabV3+ utilized atrous spatial pyramid pooling (ASPP) to capture multi-scale contextual information, while Attention U-Net incorporated attention gates to enhance feature learning. SegFormer, as a transformer-based segmentation model, employed hierarchical multi-head self-attention

mechanisms and a lightweight multi-layer perceptron (MLP) decoder to refine global feature extraction and segmentation boundaries. YOLOv5 and YOLOv10, originally designed for object detection, were adapted for segmentation by integrating a custom segmentation head while retaining their efficient real-time processing capabilities. These architecture-specific differences were preserved while ensuring identical training settings to maintain a fair comparative evaluation.

The performance of AMAY-Net was evaluated using the following metrics:

- Precision: the proportion of correctly predicted positive pixels (true positives) out of all predicted positive pixels, calculated as:

$$\text{Precision} = \frac{TP}{TP + FP}. \tag{24}$$

- Recall: the proportion of correctly predicted positive pixels out of all actual positive pixels, calculated as:

$$\text{Recall} = \frac{TP}{TP + FN}. \tag{25}$$

- F1-score: the harmonic mean of Precision and Recall, calculated as:

$$\text{F1-score} = 2 \times \frac{\text{Precision} \times \text{Recall}}{\text{Precision} + \text{Recall}}. \tag{26}$$

- Intersection* over Union (IoU): measures the overlap between the predicted segmentation mask and the ground truth, calculated as:

$$\text{IoU} = \frac{\text{Intersection}^* \ (TP)}{\text{Union} \ (TP \ + \ FP \ + \ FN)}. \tag{27}$$

## Experimental results

Table 1 presents a comprehensive comparison of AMAY-Net with several state-of-the-art segmentation models, including U-Net, DeepLabV3+, Attention U-Net, YOLOv5, YOLOv10, and SegFormer, for both liver and gallbladder segmentation. We evaluate the models using Precision, Recall, F1-score, and Intersection over Union (IoU) to capture multiple aspects of segmentation quality.

In liver segmentation, AMAY-Net achieved the highest overall performance, with a Precision of 0.951, Recall of 0.961, F1-score of 0.956, and IoU of 0.911. In contrast, U-Net attained an F1-score of 0.904 and IoU of 0.857, while DeepLabV3+, Attention U-Net, and SegFormer reached F1-scores of 0.913, 0.928, and 0.930, respectively. AMAY-Net's advantage stems from its multi-scale feature extraction and dual attention modules, which allow for better delineation of liver boundaries even in complex backgrounds. Furthermore, the adaptive class-balancing loss helped maintain high segmentation accuracy by mitigating class imbalance.

For gallbladder segmentation, AMAY-Net again led the performance metrics, achieving a Precision of 0.952, Recall of 0.882, F1-score of 0.916, and IoU of 0.826. U-Net, DeepLabV3+, and YOLOv5 reported F1-scores of 0.861, 0.862, and 0.873, respectively. Attention U-Net, YOLOv10, and SegFormer performed slightly better, with F1-scores of

**Table 1 Performance comparison of AMAY-Net and other models on liver and gallbladder segmentation.**

| Class model | Liver | | | | Gallbladder | | | |
|---|---|---|---|---|---|---|---|---|
| | Precision | Recall | F1-score | IoU | Precision | Recall | F1-score | IoU |
| U-Net | 0.913 | 0.905 | 0.904 | 0.857 | 0.889 | 0.832 | 0.861 | 0.781 |
| DeepLabV3+ | 0.924 | 0.902 | 0.913 | 0.862 | 0.881 | 0.841 | 0.862 | 0.782 |
| YOLOv5 | 0.931 | 0.911 | 0.922 | 0.873 | 0.896 | 0.851 | 0.873 | 0.796 |
| Attention U-Net | 0.941 | 0.916 | 0.928 | 0.886 | 0.921 | 0.861 | 0.890 | 0.811 |
| YOLOv10 | 0.944 | 0.931 | 0.938 | 0.893 | 0.931 | 0.871 | 0.898 | 0.817 |
| SegFormer | 0.939 | 0.922 | 0.930 | 0.887 | 0.926 | 0.866 | 0.893 | 0.813 |
| AMAY-Net | 0.951 | 0.961 | 0.956 | 0.911 | 0.952 | 0.882 | 0.916 | 0.826 |

0.890, 0.898, and 0.893. However, AMAY-Net's adaptive class-balancing strategy significantly improved segmentation accuracy of small and often underrepresented structures like the gallbladder, allowing the model to maintain high precision without sacrificing recall.

Compared to models like DeepLabV3+ and SegFormer, AMAY-Net strikes an effective balance between segmentation quality and inference efficiency. DeepLabV3+ uses atrous convolutions for multi-scale context but at the cost of slower performance, while SegFormer achieves strong global representation *via* transformers but suffers from high latency. AMAY-Net mitigates these issues through its efficient detection-based backbone and streamlined attention design, enabling it to operate in real-time environments without compromising accuracy.

Figure 3 illustrates the segmentation performance of AMAY-Net across 13 different classes using four key metrics: Precision, Recall, F1-score, and IoU. As shown in the figure, AMAY-Net performs exceptionally well in segmenting large anatomical structures like the liver and gallbladder with F1-scores of 0.956 and 0.916, respectively. These results are further supported by the high IoU values for these classes, indicating a precise overlap between the predicted segmentation and ground truth. For large background classes, such as Black Background, the model achieves near-perfect segmentation performance, with all four metrics close to 1.0. This confirms that AMAY-Net is highly accurate in distinguishing background from foreground objects during laparoscopic procedures.

To quantitatively evaluate the real-time segmentation capability of AMAY-Net, we measured the inference speed on modest hardware consisting of an Intel Xeon CPU, 64 GB RAM, and an NVIDIA GTX 1660 GPU. The results, summarized in Table 2, show that AMAY-Net achieves an inference speed of approximately 38 frames per second (FPS), clearly meeting and surpassing the typical real-time segmentation threshold of 25 FPS required in surgical contexts.

## Analysis

Figure 4 displays the segmentation results generated by AMAY-Net compared to the ground truth annotations for several laparoscopic cholecystectomy images. The top row

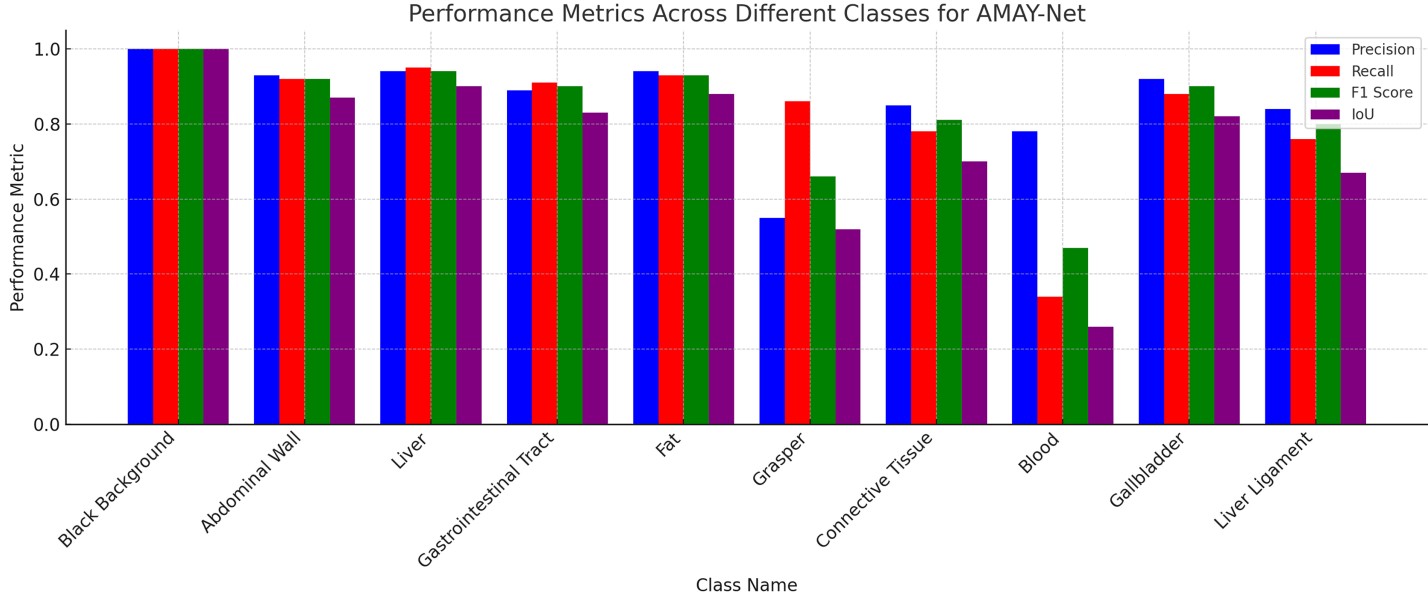

**Figure 3 Performance for each class in CholecSeg8k dataset.**

**Table 2 Inference speed comparison on modest hardware (NVIDIA GTX 1660 GPU).** Inference speeds are approximate values based on standard implementations of these models under identical hardware conditions.

| Model | Inference speed (FPS) |
| --- | --- |
| U-Net | 19 |
| DeepLabV3+ | 14 |
| YOLOv5 | 42 |
| Attention U-Net | 17 |
| YOLOv10 | 36 |
| SegFormer | 15 |
| AMAY-Net (Proposed) | 38 |

shows the ground truth masks, while the bottom row presents the corresponding predicted segmentation masks produced by AMAY-Net. The visual comparison between the predicted masks and the ground truth allows for an in-depth evaluation of the model's performance.

In the task of segmenting large anatomical structures such as the liver and abdominal wall, AMAY-Net performs exceptionally well. The predicted segmentation closely aligns with the ground truth, with most regions showing near-perfect overlap. This indicates that the model's multi-scale feature extraction module effectively captures the spatial information of larger structures, allowing it to accurately delineate the liver and abdominal wall. For smaller structures, such as the gallbladder and blood vessels, AMAY-Net demonstrates reasonably stable performance. The gallbladder, highlighted in green in the segmentation masks, is well separated from the surrounding background, accurately reflecting its boundaries in most images. However, there are a few instances where the

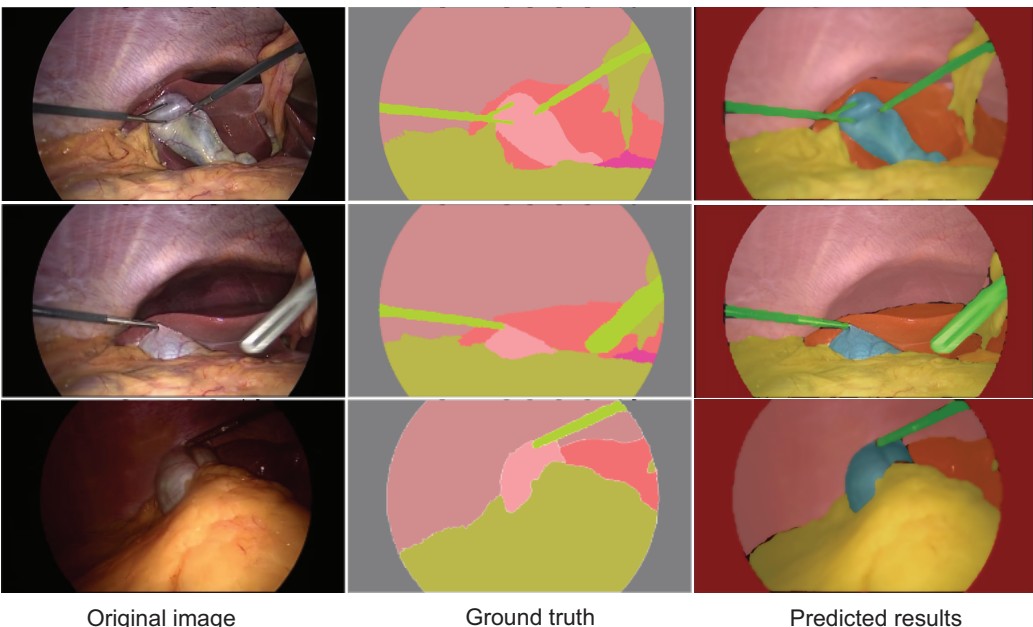

|  Original image | Ground truth | Predicted results |

**Figure 4 Qualitative segmentation results illustrating AMAY-Net's performance in CholecSeg8k Dataset.** Left column: original laparoscopic images. Middle column: corresponding ground truth masks. Right column: predicted segmentation results by AMAY-Net. In the masks, red represents the liver, and green represents the gallbladder. The images shown were randomly selected from the test dataset to transparently represent typical segmentation performance.

edges of the gallbladder appear slightly blurred, suggesting that the model encounters some difficulties when handling the finer details of smaller structures, particularly in the presence of surgical instruments or overlapping tissues. The model also performs well in segmenting surgical instruments, such as the graspers. In the images, the green-colored instruments are clearly distinguishable from the surrounding anatomical structures, such as the liver and gallbladder. Although there are minor discrepancies in a few areas, the overall precision and accuracy of the instrument segmentation are high, demonstrating the model's ability to manage surgical tool interference while simultaneously segmenting anatomical structures. However, some challenges remain, especially in areas with more complex anatomical structures. In certain images, particularly those featuring overlapping tissues and small objects like blood vessels, AMAY-Net's predictions show noticeable deviations from the ground truth. This is most evident in the segmentation of small and irregularly shaped objects, where the model struggles to capture fine details. For example, thin structures like blood vessels are often missed or inaccurately segmented, which indicates a limitation in the model's ability to handle small-object segmentation. This suggests that there is room for improvement, particularly in enhancing the model's capability to focus on small targets and refine boundaries in complex regions. Overall, the results presented in Fig. 4 show that AMAY-Net excels at segmenting anatomical structures, such as the liver and abdominal wall, and effectively handles the interference caused by surgical instruments.

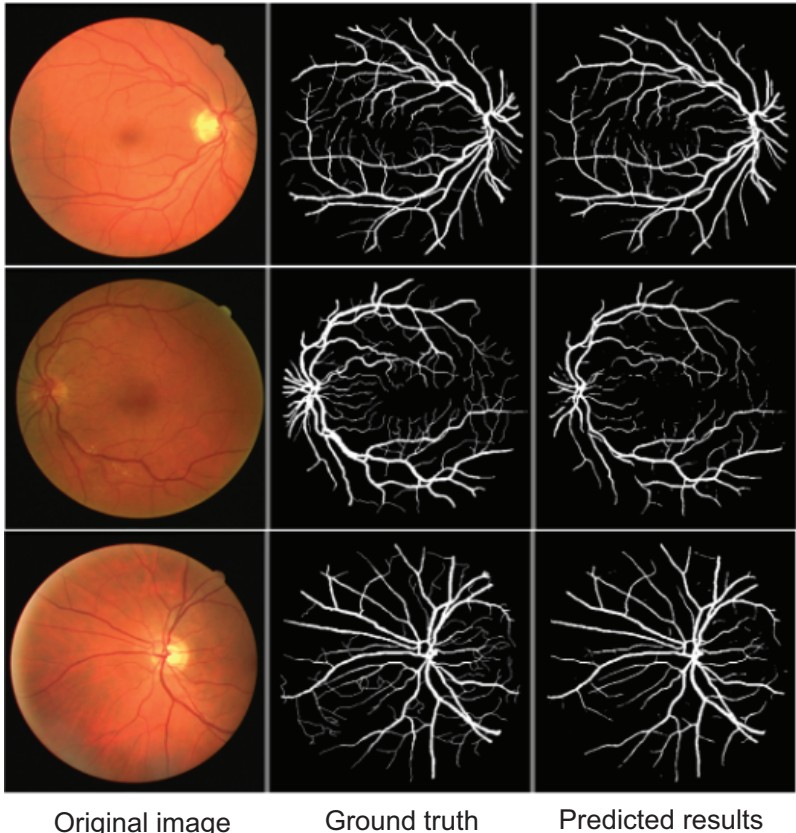

Original image          Ground truth          Predicted results

**Figure 5 Qualitative segmentation results illustrating AMAY-Net's performance in the Retina Blood Vessel Segmentation Dataset.** Left column: original laparoscopic images. Middle column: corresponding ground truth masks. Right column: predicted segmentation results by AMAY-Net. In the masks, white represents the Retina Blood Vessel. The images shown were randomly selected from the test dataset to transparently represent typical segmentation performance.

To further demonstrate AMAY-Net's ability to accurately segment smaller anatomical structures, we conducted additional validation experiments using the publicly available Retina Blood Vessel Segmentation dataset (*Staal et al., 2004*). This dataset consists of retinal fundus images annotated with pixel-level labels of fine-grained blood vessel structures, providing a representative example of segmentation tasks involving small, complex anatomical details. The qualitative results shown in Fig. 5 illustrate AMAY-Net's capability in effectively segmenting the detailed vascular structures from retinal images. The segmentation predictions closely match the ground truth annotations, highlighting AMAY-Net's potential generalizability and robustness in handling challenging segmentation tasks beyond its original laparoscopic surgical context.

## Ablation study

To demonstrate the effectiveness of the key components in AMAY-Net, we conducted an ablation study by progressively removing or replacing certain modules and observing their impact on the model's performance. Specifically, we tested the contribution of three major components: the multi-scale feature extraction module, the dual attention mechanisms

**Table 3** Ablation study on AMAY-Net components for liver and gallbladder segmentation.

| Model | Liver | | | | Gallbladder | | | |
|---|---|---|---|---|---|---|---|---|
| | Precision | Recall | F1-score | IoU | Precision | Recall | F1-score | IoU |
| Full AMAY-Net | 0.951 | 0.961 | 0.956 | 0.911 | 0.952 | 0.882 | 0.916 | 0.826 |
| w/o Multi-Scale Features | 0.926 | 0.921 | 0.923 | 0.861 | 0.911 | 0.851 | 0.879 | 0.781 |
| w/o Dual Attention Mechanism | 0.931 | 0.912 | 0.921 | 0.872 | 0.921 | 0.861 | 0.890 | 0.801 |
| w/o Class-Balancing Loss | 0.921 | 0.902 | 0.911 | 0.851 | 0.901 | 0.832 | 0.865 | 0.771 |

(spatial and channel attention), and the adaptive class-balancing loss function. The results of these ablation experiments are shown in Table 3.

The ablation study results shown in Table 3 highlight the contribution of each major component in AMAY-Net. Removing the multi-scale feature extraction module leads to a noticeable performance drop. For liver segmentation, the F1-score decreases from 0.956 to 0.923, and the IoU drops from 0.911 to 0.861. This confirms the importance of multi-scale features in capturing both global context and local anatomical detail. Excluding the dual attention mechanism also degrades performance, particularly in the gallbladder segmentation task. The F1-score for gallbladder segmentation falls from 0.916 to 0.890, and the IoU from 0.826 to 0.801, indicating that attention mechanisms play a key role in enhancing focus on relevant spatial regions and feature channels, especially for small or occluded structures. Finally, when the adaptive class-balancing loss is removed, the model exhibits reduced robustness to class imbalance. Gallbladder segmentation performance drops to an F1-score of 0.865 and an IoU of 0.771, suggesting the loss function's vital role in improving representation of underrepresented classes. This component is particularly beneficial for small anatomical targets, helping the model maintain high recall without sacrificing precision.

## CONCLUSION

In this work, we introduced AMAY-Net, a novel architecture designed for efficient and accurate segmentation in laparoscopic surgery images, specifically optimized for segmenting critical anatomical structures such as the liver and gallbladder. These structures were selected primarily due to their clinical significance, visibility during laparoscopic cholecystectomy, and the availability of reliable annotation data, providing a robust basis for validation. AMAY-Net combines YOLO's real-time detection capability with pixel-level segmentation through multi-scale feature extraction, attention mechanisms, and an adaptive class-balancing loss function. Experimental evaluations demonstrate that AMAY-Net not only achieves superior segmentation accuracy compared to established methods such as U-Net, DeepLabV3+, and Attention U-Net but also meets real-time performance requirements, achieving an inference speed of approximately 38 FPS on modest hardware (NVIDIA GTX 1660 GPU).

Despite these promising results, AMAY-Net currently faces limitations in segmenting very small and irregular anatomical structures such as the cystic duct and hepatic veins. The segmentation accuracy for these smaller structures, although improved by the

adaptive class-balancing loss function, still requires further enhancement. Future research will focus on refining attention mechanisms to capture fine-grained details and exploring additional training strategies, such as targeted data augmentation and self-supervised learning, to enhance performance on smaller, challenging structures. We also plan to further evaluate and generalize the model by incorporating other publicly available datasets with fine-scale anatomical structures.

### Funding
The authors received no funding for this work.

### Competing Interests
The authors declare that they have no competing interests.

### Author Contributions
- Yuyang Zhou conceived and designed the experiments, performed the experiments, analyzed the data, performed the computation work, prepared figures and/or tables, authored or reviewed drafts of the article, and approved the final draft.
- Yulai You performed the experiments, analyzed the data, prepared figures and/or tables, and approved the final draft.
- Xiaokai Tan performed the computation work, authored or reviewed drafts of the article, and approved the final draft.
- Juncheng Tang conceived and designed the experiments, authored or reviewed drafts of the article, and approved the final draft.

### Data Availability
The data is available at Kaggle:
- https://www.kaggle.com/datasets/abdallahwagih/retina-blood-vessel.
- https://www.kaggle.com/datasets/newslab/cholecseg8k.

### Supplemental Information
Supplemental information for this article can be found online at http://dx.doi.org/10.7717/peerj-cs.2961#supplemental-information.

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
