# Peer review of "AMAY-Net: adaptive multi-scale attention YOLO network for liver and gallbladder segmentation in laparoscopic cholecystectomy"

_PeerJ Computer Science, doi:10.7717/peerj-cs.2961_

## Round 0.1 · original submission · Major Revisions

Dear Authors,
Your paper has been revised. Given the reviewers' criticism, it needs major revisions before being considered for publication in PEERJ Computer Science. More precisely, the following points must be faced in the revised version of your paper:

1) The methodology section should include a comparison with other segmentation techniques. This could help justify your model's choice and demonstrate its advantages in liver cancer diagnosis.
2) Fig. 1 should clearly visualize the overview of the AMAY-Net. Furthermore, because the introduction section mentions the capacity for real-time image segmentation as a major need/motivation for the present work, it is necessary to indicate the inference times on modest hardware and whether the proposed model is capable of real-time segmentation.

Reviewer 1 ·

Basic reporting

Clear, unambiguous, professional English
language used throughout.

The language which is used throughout the journal paper is clear, unambiguous and also professional.
Intro & background to show context is satisfactory.
Literature review can be made stronger by adding few more papers.
Structure conforms to PeerJ standards,
discipline norm, or improved for clarity.
Figures are relevant, high quality, well
labelled & described.

Experimental design

Original primary research within Aims and Scope of the journal.

Research question well defined, relevant & meaningful. It is stated how research fills an identified knowledge gap.

Rigorous investigation performed to a high technical & ethical standard.

The paper is written with sufficient methods described with sufficient detail

Validity of the findings

No Comment

Additional comments

Additional information about the results can be highlighted.

·

Basic reporting

1. The motivations of the manuscript could be further presented visually.

2. The overview of the AMAY-Net should be visualized clearly in Fig.1.

Experimental design

1. The YOLO-based method should be compared in the experiment.

Validity of the findings

1. The qualitative results should be presented. For instance, the visualization of Liver and Gallbladder Segmentation results.

Reviewer 4 ·

Basic reporting

see below

Experimental design

see below

Validity of the findings

see below

Additional comments

This work describes the performance of a new YOLO-based network for liver and gallbladder semantic segmentation in cholecystectomy imaging. There are several shortcomings in terms or methodology and interpretation of results that need to be addressed:

Major comments:
* Introduction: "Accurate segmentation of critical anatomical structures such as the liver and gallbladder is essential for the success and safety of this surgery" - This sentence does not make any medical sense; there is currently no demonstrated safety benefit (i.e., reduced complication rates, reduced rate of bile duct injury) of intraoperative real-time anatomy segmentation whatsoever.
* "Semantic segmentation has become a critical component of medical imaging, driven by the need to accurately segment anatomical structures from complex, high-resolution images" - similar problem here - the sentence is medically incorrect, there is no true clinical "need" "to accurately segment anatomical structures".
* Your description of the dataset splits reads like you randomly selected 80%/10%/10% from the segmented images. This introduces the risk of data leakage if images from the same surgery are included in training and test sets. Please make sure to split your dataset along surgery lines (i.e., data from one surgery can only be included in either training, validation, or test set).
* You describe the implementation of AMAY-Net in a lot of detail, but the implementation of the models you compare your results with (DeepLabV3+, Attention U-Net, U-Net) is not described well. Please add details to make sure your results are reproducible.
* Why did you select liver and gallbladder as your main segmented structures of interest that you describe in Table 1? This choice seems a bit puzzling because you described the importance of segmenting smaller structures well as one of the main motivations for the work - but then show results for liver and gallbladder, two large structures.
* Results: "critical structures like the liver and gallbladder" - these structures are typically easy to identify for surgeons. Could you please display your results for cystic duct and hepatic vein segmentation as well? There are also other anatomy segmentation datasets that truly contain small structures (i.e., ureter, blood vessels) - DOI 10.1038/s41597-022-01719-2.

* Overall, I feel that your results do not match the motivation outlined in the introductory sections of your manuscript (i.e., balancing segmentation performance for structures of varying sizes).
* In FIgure 4, please label in the figure what the ground truth masks are and what the predicted results are, and which color indicates what segment. How did you select the images that you display here? Please make sure to be transparent in your results presentation.
* In the introduction, you mention the capacity for real-time image segmentation as a major need/motivation for your work. What were the inference times on modest hardware? Is your model capable of real-time segmentation?
* I was missing a meaningful discussion of your results. What are the limitations of your work? How does your work compare with results described on DeepLabV3-based and SegFormer-based anatomy segmentation (i.e. described in DOI: 10.1097/JS9.0000000000000595)?


Minor comments:
* The citations are currently formatted in a style that makes it hard to follow, please add brackets or superscript numbers and/or remove author names.
* Bold question mark in the "related work" section, unclear what is meant
* Both section 3 and section 4 are titled "Experimental Results"
* Figure 1 and Figure 2 are not cited in the text

---

## Round 0.2 · Major Revisions

Dear Authors,

Your paper has been revised. Based on the reviewer's report, major revisions are needed before it is accepted for publication in PEERJ Computer Science. More precisely:

1) You must add the exact surgery IDs from CholecSeg8k that were part of your database splitting for the reproducibility of the numerical experiments described in your work.

Reviewer 4 ·

Basic reporting

see below

Experimental design

see below

Validity of the findings

see below

Additional comments

Thank you for submitting the revised manuscript version. I feel the manuscript has improved a lot in terms of depth and readability. I have a few remaining concerns:

Major:
* I still have concerns about the division of the dataset. While you have changed the wording to clarify that the dataset was split (80/10/10) along surgery lines to avoid data leakage, I am still missing details on the exact surgery IDs from CholecSeg8k that were part of each of the splits. Please add these for reproducibility.

Minor:
* Please reference DOI: 10.1097/JS9.0000000000000595 in your manuscript - you are making a number of implicit acknowledgments of this work, but it is not cited yet
* I think the reference "Carstens et al." (line 520) is misplaced...

---

## Round 0.3 · Major Revisions

Dear Authors,
Your paper has been revised. It needs major revisions before being considered for publication in PEERJ Computer Science. More precisely, as noted by Reviewer 4:

1) Based on your explanation of the dataset splits, the test set was not truly independent, given that frames from the same video were used in the validation set. Therefore, the test set is unseen at the patient/surgery level. From a clinical standpoint, the test set should always be genuinely independent of patients/surgeries. Please re-run the experiments with dataset splits along patient/surgery lines.

Reviewer 4 ·

Basic reporting

see below

Experimental design

see below

Validity of the findings

see below

Additional comments

Thank you for addressing my remaining comments. I disagree with publication of this work in its current form. Based on your explanation of the dataset splits, the test set was not truly independent, given that frames from the same video were used in the validation set. Therefore, the test set is not truly unseen on the patient/surgery level. From a clinical standpoint, the test set should always be truly independent patients/surgeries. Please re-run the experiments with dataset splits along patient/surgery lines.

---

## Round 0.4 · accepted · Accept

Dear Authors,

Your paper has been revised. It has been accepted for publication in PEERJ Computer Science. Thank you for your fine contribution.

Reviewer 4 ·

Basic reporting

See below

Experimental design

See below

Validity of the findings

See below

Additional comments

Thank you very much for revising your work and redoing your experiments with splits along patient lines. Congratulations on a great paper!